

# Soil aggregates indirectly influence litter carbon storage and release through soil pH in the highly alkaline soils of north China

Chao Yang[1,2], Jingjing Li[1] and Yingjun Zhang[1,3]

[1] College of Grassland Science and Technology, China Agricultural University, Beijing, China
[2] College of Grassland Science, Qingdao Agricultural University, Qingdao, China
[3] Key Laboratory of Grassland Management and Rational Utilization, Ministry of Agriculture, Beijing, China

## ABSTRACT

**Background:** Soil aggregate-size classes, structural units of soil, are the important factors regulating soil organic carbon (SOC) turnover. However, the processes of litter C mineralization and storage in different aggregates-size classes are poorly understood, especially in the highly alkaline soils of north China. Here, we ask how four different aggregate sizes influence rates of C release ($C_r$) and SOC storage ($C_s$) in response to three types of plant litter added to an un-grazed natural grassland.

**Methods:** Highly alkaline soil samples were separated into four dry aggregate classes of different sizes (2–4, 1–2, 0.25–1, and <0.25 mm). Three types of dry dead plant litter (leaf, stem, and all standing dead aboveground litter) of *Leymus chinensis* were added to each of the four aggregate class samples. Litter mass loss rate, $C_r$, and $C_s$ were measured periodically during the 56-day incubation.

**Results:** The results showed that the mass loss in 1–2 mm aggregates was significantly greater than that in other size classes of soil aggregates on both day 28 and day 56. Macro-aggregates (1–2 mm) had the highest $C_r$ of all treatments, whereas 0.25–1 mm aggregates had the lowest. In addition, a significant negative relationship was found between $C_s/C_r$ and soil pH. After incubation for 28 and 56 days, the $C_s$ was also highest in the 1–2 mm aggregates, which implied that the macro-aggregates had not only a higher $CO_2$ release capacity, but also a greater litter C storage capacity than the micro-aggregates in the highly alkaline soils of north China.

# INTRODUCTION

Soils contain more than twice the carbon than the atmosphere and play an important role in the C cycle (*Davidson, Trumbore & Amundson, 2000*; *Schmidt et al., 2011*). Litter represents a major source of soil organic carbon (SOC) and generally more than 50% of net primary production is returned to the soil via decomposition of plant litter in terrestrial ecosystems (*Garcia-Palacios et al., 2013*; *Wardle et al., 2004*). Consequently, litter decomposition is a crucial step in the carbon cycle (*Schmidt et al., 2011*), especially in

Corresponding author
Yingjun Zhang, zhangyj@cau.edu.cn

grassland ecosystems, which cover 40% of the earth's land surface (*Lu et al., 2017*), and contain approximately 20% of the global SOC stock (*Schuman, Janzen & Herrick, 2002*).

The majority of studies have shown that litter decomposition usually depends on three main factors: climate factors (soil moisture, temperature) (*Wang, Zeng & Zhong, 2016*; *Zhong et al., 2017*), litter quality (i.e., its chemical composition) (*Hishinuma et al., 2017*; *Zhang et al., 2016*) and composition and activity of the soil decomposer community (*Keiser & Bradford, 2017*; *Marella et al., 2016*). Under specific climatic conditions, litter quality is an important driver of litter carbon decomposition and nutrient release (*Manzoni et al., 2010*). In general, plant litter with high nutrients and low lignin content (low C-to-Nut ratio) decays faster than litter with low nutrient and high lignin contents (*Freschet, Aerts & Cornelissen, 2012*). Litter decomposition can differ substantially between plant species or plant functional types within the same ecosystem (*Patoine et al., 2017*). Decomposition rates may also differ between different plant tissues of the same species. For example, root litter generally decays slower than leaf litter (*Fujii & Takeda, 2010*; *Ma et al., 2016*). In addition, there is now growing evidence that decomposer community composition influences litter decomposition rates over and above climate and litter quality (*Bradford et al., 2016*; *Schimel & Schaeffer, 2012*). Soil microbial processes are regulated by soil pH, which is considered to be an important factor controlling the composition of soil microbial communities (*Lauber et al., 2009*; *Rousk et al., 2010*), and thus litter decomposition.

Conceptually, aggregates are generally classified into macro-aggregates (>0.25 mm) and micro-aggregates (<0.25 mm) (*Liu et al., 2014*; *Yang, Liu & Zhang, 2017*). SOC mineralization in macro-aggregates is considered to be greater than micro-aggregates (*Fernández et al., 2010*; *Rabbi et al., 2014*), because the reduced diffusion of oxygen into micro-aggregates which leads to reduced microbial activity within the micro-aggregates (*Stamati et al., 2013*). Therefore, micro-aggregates are the main site of carbon storage because of their lower carbon release capacity and greater physical protection. However, this does not mean that soil micro-aggregates have a higher litter decomposition capacity to convert litter carbon into soil compared with macro-aggregates. The decomposition capacity of litter carbon in the two types of soil aggregates is not well understood. In addition, aggregate size significantly influences soil pH (*Jiang et al., 2013*), and it is not clear how soil pH affects litter decomposition within macro- vs. micro-aggregates.

We designed a two-factor experiment in the laboratory: one factor was litter type (leaf and stem), the other factor was soil aggregate size. The temporal changes in SOC mineralization, SOC content, and soil pH were measured. We hypothesized that: (1) soil aggregate size and soil pH are correlated and control litter decomposition, and consequently, (2) there is a threshold relationship between soil pH and litter decomposition, and (3) macro-aggregates have higher litter C concentrations than micro-aggregates, despite greater losses of carbon.

## MATERIALS AND METHODS

### Material collection

The soil in this study was collected from a natural grassland located at the Guyuan National Grassland Ecosystem Research Station in the agro-pastoral transition region of

**Table 1 Initial mean (±SE, $n$ = 3) total carbon (TC), total nitrogen (TN), and carbon to nitrogen ratio (C/N) for different soil aggregate size classes and litter types.** The proportion of the soil in each aggregate size class is also presented.

| | Soil aggregates | | | | | Litter type | | |
|---|---|---|---|---|---|---|---|---|
| | 2–4 mm | 1–2 mm | 0.25–1 mm | <0.25 mm | | Leaf | Stem | All |
| SOC (g kg$^{-1}$) | 13.27 (0.1)$^{ab}$ | 11.87 (0.2)$^{b}$ | 5.20 (0.2)$^{c}$ | 14.17 (0.3)$^{a}$ | TC (g kg$^{-1}$) | 411.96 (0.1)$^{c}$ | 424.19 (0.1)$^{a}$ | 417.81 (1.3)$^{b}$ |
| TN (g kg$^{-1}$) | 1.67 (0.03)$^{bc}$ | 1.73 (0.03)$^{b}$ | 1.07 (0.07)$^{c}$ | 1.90 (0.06)$^{a}$ | TN (g kg$^{-1}$) | 17.20 (0.1)$^{a}$ | 14.30 (0.1)$^{c}$ | 16.53 (0.3)$^{b}$ |
| C/N ratios | 7.95 (0.01)$^{a}$ | 6.86 (0.01)$^{b}$ | 4.86 (0.02)$^{c}$ | 7.46 (0.01)$^{a}$ | C/N ratios | 23.95 (0.1)$^{c}$ | 29.66 (0.1)$^{a}$ | 25.27 (0.2)$^{b}$ |
| pH | 8.24 (0.01)$^{c}$ | 8.21 (0.01)$^{c}$ | 8.45 (0.01)$^{a}$ | 8.28 (0.01)$^{b}$ | | | | |
| Proportion (%) | 12.71 (1.07)$^{c}$ | 5.76 (0.32)$^{d}$ | 38.26 (1.49)$^{b}$ | 42.09 (0.98)$^{a}$ | | | | |

Note:
Different letters in the same column indicate a significant difference at $p < 0.05$ using least-significant difference tests.

northern Hebei Province in China (41°46′N, 115°41′E, elevation 1,380 m) in May of 2018. This area is a typical temperate zone characterized by a mean annual precipitation of 430 mm and a mean annual temperature of 1.4 °C. The minimum monthly mean air temperature is −18.6 °C in January and the site reaches a maximum of 21.1 °C in July. Precipitation primarily falls during the growing season (June–August), which coincides with the highest temperatures (*Yang et al., 2019*). The site has a calcic-orthic Aridisol (highly alkaline) soil with a loamy-sand texture, and the carbonate content is about 12 g kg$^{-1}$ (*Luo et al., 2015*). The sand:silt:clay ratio is about 40:10:1, and the cation exchange capacity is around 20 cmol (+) kg$^{-1}$ (*Cai et al., 2017*; *Li et al., 2019*). Some basic characteristics for the soils in Table 1 were cited from our previous studies (*Yang, Liu & Zhang, 2017*, *2019*).

The top layer (0–15 cm) of soil was transported to the laboratory, where plant roots and leaves were carefully removed by hand, after which the soil was spread in a thin layer and air-dried. The dried soil was sieved to separate large macro-aggregates (2–4 mm), macro-aggregates (1–2 mm), meso-aggregates (0.25–1 mm), and micro-aggregates (<0.25 mm). Soil aggregates were separated into different size fractions by dry sieving in accordance with the method in *Elliott (1986)*. The undisturbed soil was shaken through four sieves (4, 2, 1, and 0.25 mm) for 2 min. We removed the >4 mm soil because there were few of these aggregates in grassland soil. Thereafter, the large macro-aggregates (2–4 mm) were collected from the two mm sieve, macro-aggregates (1–2 mm) from the one mm sieve, meso-aggregates (0.25–1 mm) from the 0.25 mm, and micro-aggregates (<0.25 mm) passed through the 0.25 mm sieve (*Yang, Liu & Zhang, 2017*). Here, the micro-aggregates will also have silt + clay particles, and we classify all these size classes as "micro-aggregates" according to *Wang, Li & Zheng (2017)*. Although air drying of soil sample is not representative of the communities that originally existed in the soil, it can represent the difference in the distribution of microbes in our incubation conditions according to *Yang, Liu & Zhang (2019)*.

In September 2018, three types of plant litter (leaf, stem, and all standing dead aboveground litter) of the dominant species *Leymus chinensis* were collected. The litter was brought to the laboratory, dried at 65 °C to constant weight and divided into

two subsamples. In order to avoid the effects of litter size on decomposition, one subsample was cut into ca. one cm long and then used for the incubation experiment. The other subsample was milled (<0.25 mm) for the analysis of chemical properties.

## Experimental design

The air-dried soil samples (200 g dry weight) of each aggregate size class (2–4, 1–2, 0.25–1, and <0.25 mm) were placed in a thin and loose layer on the bottom of 1,000 mL jars. Each aggregate size had three replicates. Three types of plant litter (three g of dry matter) were mixed with 200 g of dry soil at a 1.5% litter-soil rate in the microcosms. A no litter addition treatment was used as the control (CK). There were a total of 96 microcosms (4 aggregate sizes × 4 litter types × 3 replications × 2 sampling times). The moisture content was adjusted to 30%, that is, the maximum field water capacity of the soil in our study. Each microcosm was covered with a perforated adherent film in order to reduce humidity loss while allowing gaseous exchange. Before adding litter, the microcosms were pre-incubated under darkness for 3 days at a constant temperature of 25 °C and under a relative humidity of 90% to allow the microbial population to colonize. After the pre incubation period, the plant litter was added, and the microcosms were maintained in the dark for 56 days at 25 °C. During the 56-day incubation, the soil moisture in each microcosm was maintained consistently by weighing each microcosm every week and adding distilled water.

After 28 and 56 days of incubation, 48 microcosms were retrieved, respectively. Litter was removed from each microcosm, cleaned with water to remove adhering soil particles, dried (65 °C, 48 h), and weighed. Soil samples were air dried for SOC concentration and pH assays. SOC concentration was measured after soaking 10 g soil with 30 mL of 0.5M HCl using an auto-analyzer (TOC; Elementar, Langenselbold, Germany), and soil pH was determined after shaking soil with water (1:2.5 w/v) for 30 min.

## Soil aggregate respiration measurement

Soil respiration was measured after 1, 7, 14, 28, 42, and 56 days of incubation. In brief, small vials with five mL of 1M NaOH were placed in the incubation jars to trap $CO_2$. In addition, three incubation jars containing only NaOH were used as blanks to correct for the $CO_2$ trapped in the air inside the vessels. The soil respiration (g $CO_2$-C kg$^{-1}$ soil day$^{-1}$) was estimated by titrating two mL NaOH from each trap and two mL 1M $BaCl_2$ (1:1) with 0.1M HCl and phenolphthalein indicator (1% w/v in ethanol) using a Digital Burette continuous E (VITLAB, Grossostheim, Germany) according to *Butterly et al. (2016)*.

To measure the litter carbon storage capacity of the soil aggregates, we used the following equation based on *Helfrich et al. (2008)*:

$$C_s/C_r = \frac{SOC_L - SOC_{CK}}{(CO_2\text{-}C)_L - (CO_2\text{-}C)_{CK}}$$

where $C_s$ in the SOC storage, $C_r$ is the $CO_2$-C release, L is the SOC concentration, $CO_2$-C is the amount released under litter addition, and CK indicates no litter addition. A ratio of one indicates that the carbon storage and release capacity of the soil aggregate are the same.

**Table 2 The two-way analysis of variance (ANOVA) used to test the effects of aggregates size and litter type on litter mass loss, $CO_2$ release rate, SOC concentration, and $C_s/C_r$ ratio.**

|        | Sources | Litter mass loss (%) | | | $CO_2$ release rate (g C kg$^{-1}$ aggregate) | | | SOC (g kg$^{-1}$) | | | $C_s/C_r$ | | |
|--------|---------|------|------|------|------|------|------|------|------|------|------|------|------|
|        |         | df | F | p | df | F | p | df | F | p | df | F | p |
| Day 28 | Size | 3 | 1,684.2 | 0.001 | 3 | 401.5 | 0.001 | 3 | 421.8 | 0.001 | 3 | 17.4 | 0.001 |
|        | Type | 2 | 152.6 | 0.001 | 3 | 5,860.9 | 0.001 | 3 | 26.8 | 0.001 | 2 | 4.6 | 0.020 |
|        | Size × Type | 6 | 6.5 | 0.001 | 9 | 10.0 | 0.001 | 9 | 3.1 | 0.008 | 6 | 1.7 | ns |
| Day 56 | Size | 3 | 3,095.6 | 0.001 | 3 | 371.7 | 0.001 | 3 | 1,477.9 | 0.001 | 3 | 44.8 | 0.001 |
|        | Type | 2 | 152.6 | 0.001 | 3 | 2,662.6 | 0.001 | 3 | 51.7 | 0.001 | 2 | 0.1 | ns |
|        | Size × Type | 6 | 6.5 | 0.001 | 9 | 13.4 | 0.001 | 9 | 13.7 | 0.001 | 6 | 1.9 | ns |

Notes:
$p < 0.05$ indicates a significant difference.
ns, not significant.

## Statistical analysis

We evaluated the normality of the data from each microcosm before analysis using a one-sample Kolmogorov–Smirnov test, which revealed that all variables followed a normal distribution. Two-way analysis of variance was used to test the effects of soil aggregate size and litter type on the $CO_2$ release rate, SOC concentration, $C_s/C_r$, and litter mass loss. The level of significance was defined at $p < 0.05$ using the least significant difference in SPSS (ver. 19.0; IBM, Armonk, NY, USA). The regression analyses and figures were drawn with SigmaPlot (ver. 12.5; Systat Software, Inc., San Jose, CA, USA).

## RESULTS

### Initial soil aggregates and litter chemistry

The initial chemical composition of all three litters differed substantially (Table 1). Litter C concentrations were higher in stem and decreased by 2.9% in leaf compared with that in stem. Litter N concentrations were lower in stem litter and increased by 20.3% in leaf compared with that in stem. In addition, the C/N ratios were higher in stem litter and decreased by 19.3% in leaf compared with that in stem. The C, N concentrations and C/N ratios for all standing dead aboveground litter were intermediate. The 2–4, 1–2, and 0.25–1 mm exhibited decreased SOC concentrations by 6.4%, 16.2%, and 63.3% compared with that in <0.25 mm, and decreased N concentrations by 12.1%, 8.9%, and 43.7% compared with that in <0.25 mm (Table 1). C/N ratios were higher in 2–4 mm and decreased by 13.7%, 38.8%, and 6.2% in 1–2, 0.25–1, and <0.25 mm compared with that in 2–4 mm. Soil pH was higher in 0.25–1 mm compared with that in 1–2 mm. In addition, the proportions of each aggregate size in the soil were in the order (<0.25 mm) > (0.25–1 mm) > (2–4 mm) > (1–2 mm) ($p < 0.05$).

### Litter mass loss, soil $CO_2$ release, and SOC storage

Soil aggregate size, litter type and their interaction significantly influenced litter mass loss, $CO_2$ release, and SOC concentration (Table 2). In three litter addition treatments, the litter mass loss in the 1–2 and 2–4 mm aggregate was significantly higher than that in

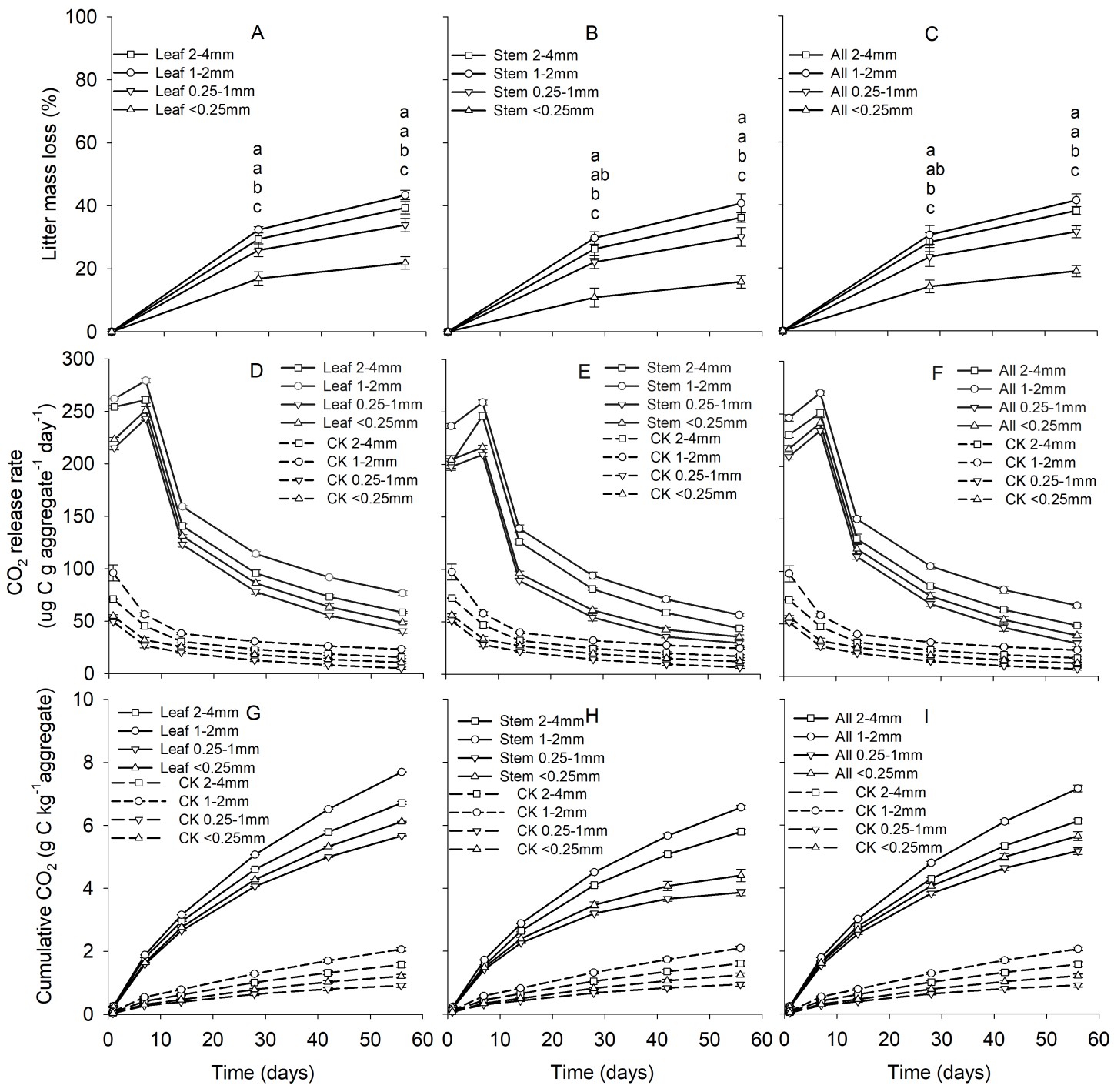

**Figure 1** Patterns of litter mass loss (A–C), $CO_2$ release rate (ug C g aggregate$^{-1}$ day$^{-1}$) (D–F), and the cumulative $CO_2$ (g C kg aggregate$^{-1}$) (G–I) over 56 days from four aggregates size under three litter addition treatments.

0.25–1 and <0.25 mm soil aggregate on both day 28 and day 56 (Figs. 1A–1C, $p < 0.05$). In all three litter addition treatments, the $CO_2$ release rate followed a similar trend with a rapid increase in the first 7 days and then slowed from the initial rapid rate during the remaining decomposition for all four aggregate sizes (Figs. 1D–1F). The 1–2 mm

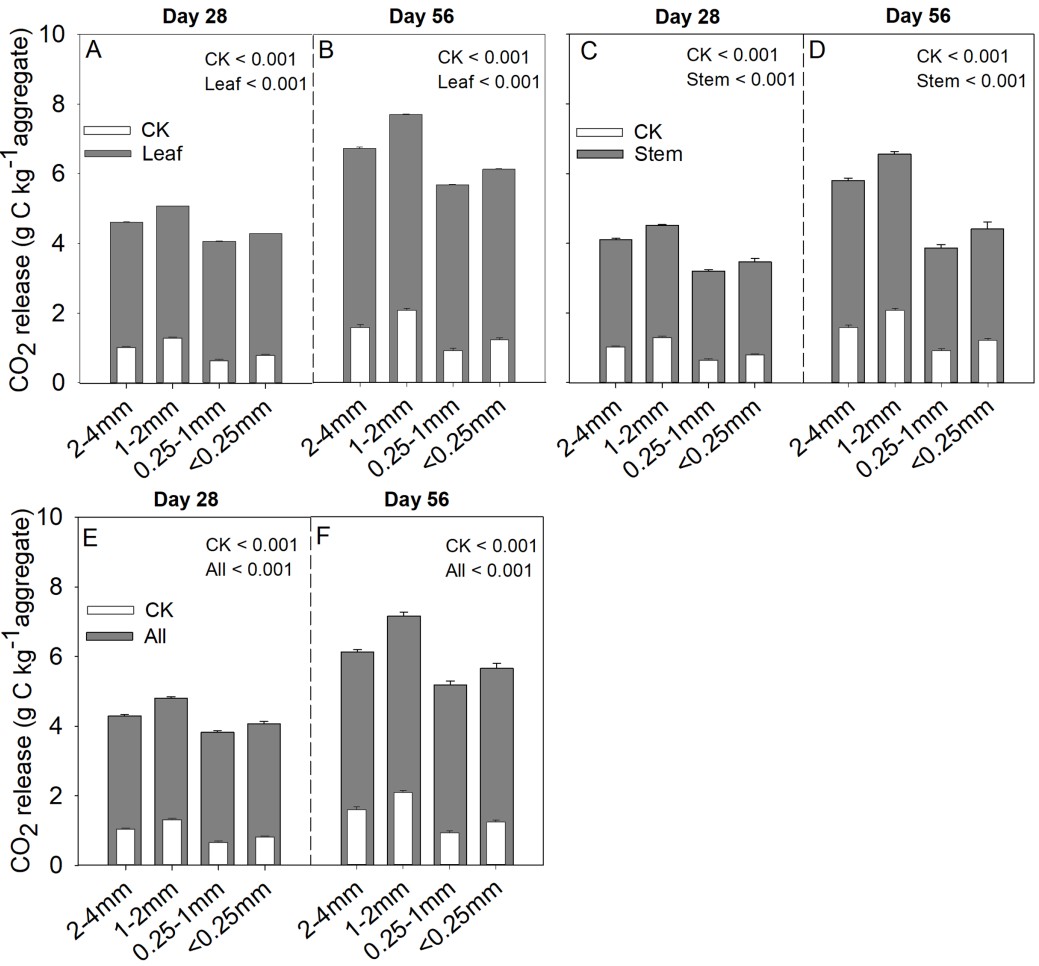

**Figure 2 Cumulative $CO_2$ (g C kg aggregate$^{-1}$) production over 28 and 56 days in the four soil aggregate size classes under leaf addition (A, B), stem addition (C, D), and leaf + stem addition (E, F) treatments.** $p < 0.05$ indicates a significant difference between four soil aggregates. The error bars show the SD of the means for $n = 3$.

aggregates had the highest $CO_2$ release rate across all treatments and the 0.25–1 mm aggregates the lowest. Correspondingly, the 1–2 mm aggregates had the highest cumulative $CO_2$ for all treatments, and the 0.25–1 mm aggregates the lowest (Figs. 1G–1I), and this difference was significant on day 28 and day 56 (Fig. 2). After incubation for 28 and 56 days, the SOC concentrations were highest in the 1–2 mm aggregate fraction (Fig. 3). In addition, $C_r$ and $C_s$ were also highest in the 1–2 mm aggregate fraction (Figs. S1 and S2).

## Correlation analysis between soil pH and carbon storage capacity

Soil aggregate size significantly influence $C_s/C_r$ ratios on both day 28 and day 56 (Table 2). The 1–2 mm aggregates had the highest $C_s/C_r$ ratios in the three litter addition treatments and for the two incubation periods. The $C_s/C_r$ ratios for all treatments decreased from day 28 to day 56 (Fig. 4). For example, the $C_s/C_r$ ratios for the 1–2 mm aggregates were 1.6–4.1 times higher than those of other aggregates on day 28 in

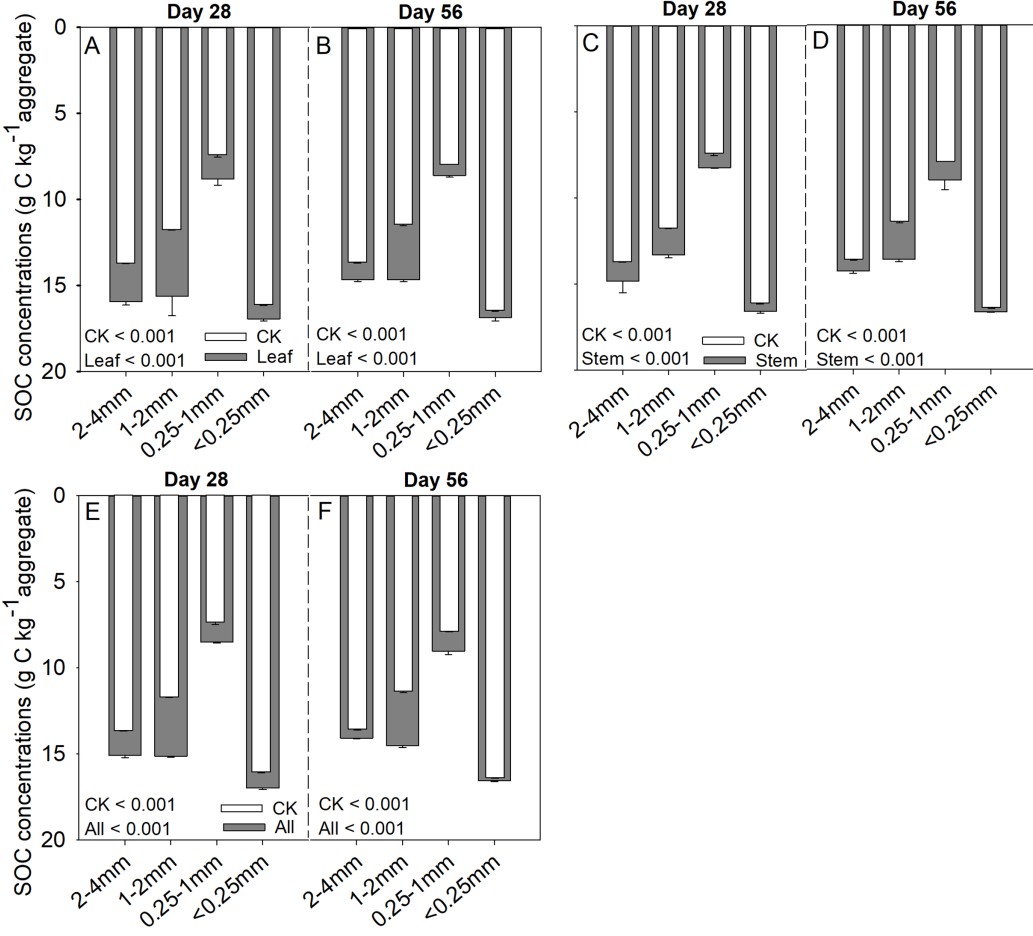

**Figure 3 SOC (g C kg aggregate$^{-1}$) concentrations over 28 and 56 days in the four soil aggregate size classes under leaf addition (A, B), stem addition (C, D), and leaf + stem addition (E, F) treatments.** $p < 0.05$ indicates a significant difference between four soil aggregates. The error bars show the SD of the means for $n = 3$.

the leaf addition treatment (Fig. 4A). Similarly, the $C_s/C_r$ ratios of the 1–2 mm aggregates were 1.6–3.8 times higher than those of other aggregates on day 28 in the stem addition treatment (Fig. 4C), and the $C_s/C_r$ ratios of the 1–2 mm aggregates were 2.3–3.6 times higher than those of other aggregates on day 28 in leaf and stem addition treatment (Fig. 4E). Litter application decreased the soil pH for all aggregate class sizes compared with the same soils without litter on days 28 and 56. In addition, the 0.25–1 mm aggregate size had the highest pH of the three litter addition treatments and two incubation periods (Fig. 5), and the relative change in soil pH was higher in 0.25–1 mm aggregate compare with that in other aggregates (Fig. S3).

Regression analysis of soil pH with $CO_2$ release, SOC concentration, $C_s/C_r$ ratios, and litter mass loss is depicted in Fig. 6. The $CO_2$ release rate of soil was negatively correlated with soil pH though a quadratic trend (Fig. 6A, $R^2 = 0.39$, $p = 0.0003$), and the inflection point was 8.3 for soil pH, and 332 for the $CO_2$ release rate. A significant trend of decreasing SOC concentration with increasing soil pH was observed (Fig. 6B, $R^2 = 0.83$,

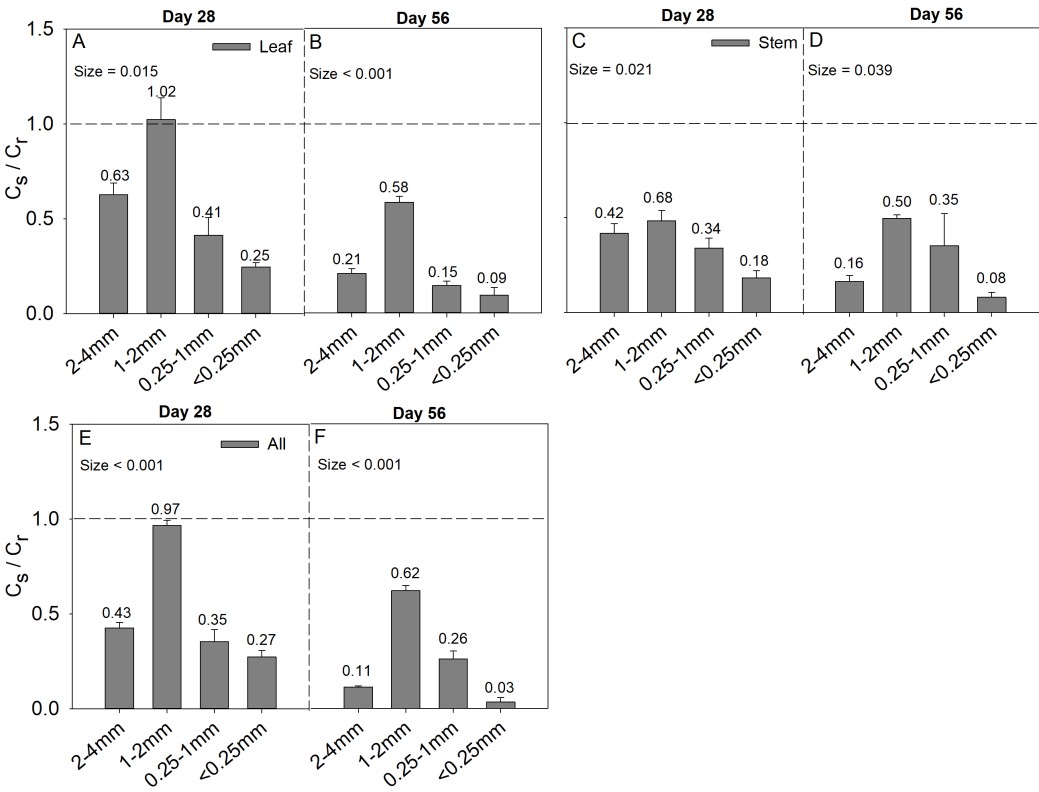

**Figure 4 $C_s/C_r$ ratio after 28 and 56 days for the four soil aggregate size classes under leaf addition (A, B), stem addition (C, D), and leaf + stem addition (E, F) treatments.** $p < 0.05$ indicates a significant difference between four soil aggregates. The error bars show the SE of the means for $n = 3$.

$p < 0.0001$). In addition, significant negative quadratic relationships were observed between soil pH and the $C_s/C_r$ ratios, and the vertex coordinates of the quadratic function were 8.37 for soil pH and 0.56 for the $C_s/C_r$ ratios (Fig. 6C, $R^2 = 0.21$, $p = 0.02$). The litter mass loss showed a negative quadratic relationship with soil pH, and the vertex coordinates of the quadratic function were 8.43 for soil pH and 47.7% for litter mass loss (Fig. 6D, $R^2 = 0.22$, $p = 0.01$).

# DISCUSSION

## Soil aggregates regulate litter mass loss, soil $CO_2$ release, and SOC storage

Numerous studies have shown that plant litter decomposition is often correlated with the chemical composition of the litters, such as N content, C/N ratio, and lignin/N ratio (*Aerts, 1997*; *Chen et al., 2019*; *Steinwandter et al., 2019*; *Yang & Chen, 2009*). Plant litter with high nutrients and low lignin content decay faster than litter with low nutrients and high lignin content (*Freschet, Aerts & Cornelissen, 2012*; *Zhang et al., 2008*). The results of the current study partly support this statement: High-quality plant litter (leaf) has a high litter mass loss rate, and soil $CO_2$ release and SOC storage also higher compared with low-quality plant litter (stem). In addition to litter quality, litter mass loss

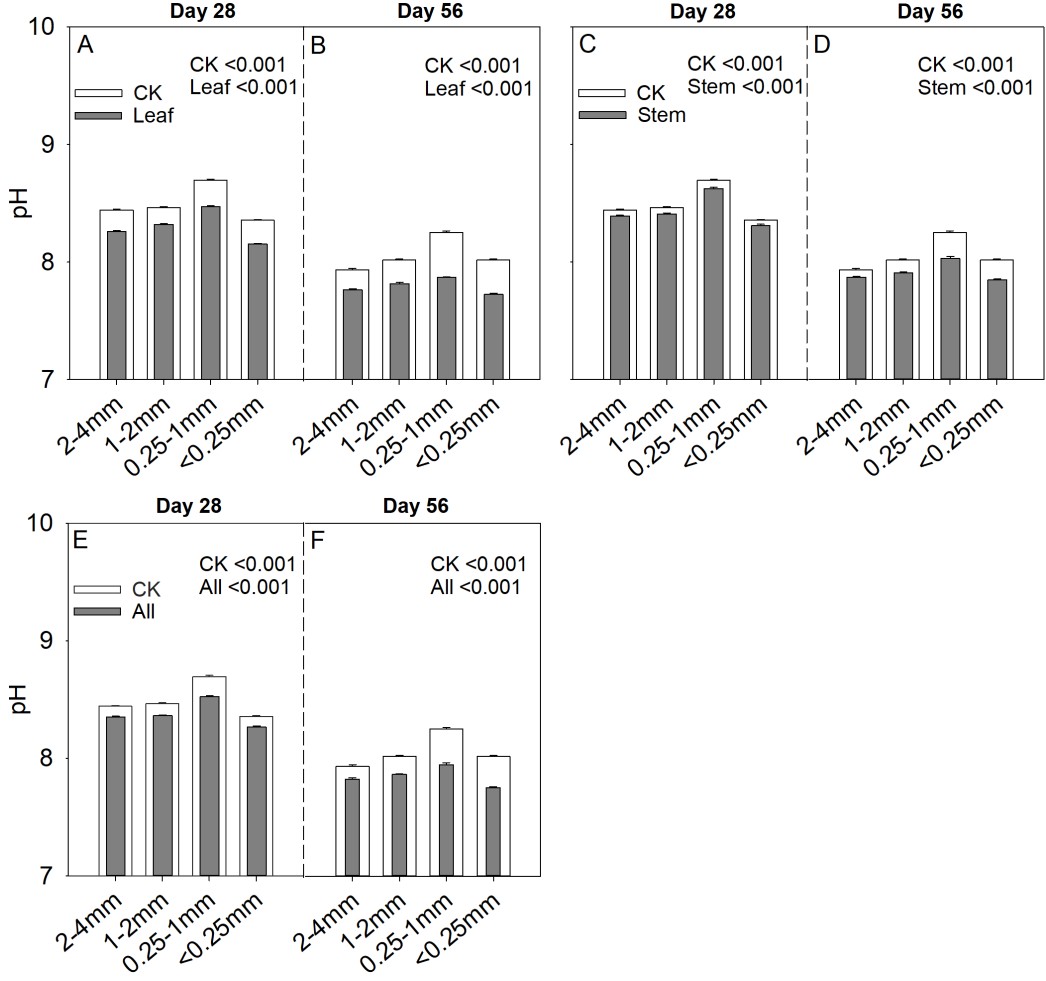

**Figure 5 Soil pH after 28 and 56 days for the four soil aggregate size classes under leaf addition (A, B), stem addition (C, D), and leaf + stem addition (E, F) treatments.** $p < 0.05$ indicates a significant difference between four soil aggregates. The error bars show the SE of the means for $n = 3$.

during incubation was also significantly affected by the size of aggregates. In general, significantly higher rates of litter mass loss occurred in soil macro-aggregates than in soil micro-aggregates (*Jha et al., 2012*), which were also close to our observation on the decreasing rates of litter mass loss in soil micro-aggregates. This may be because of greater contact with air filled pores and microorganisms in macro-aggregates as compared to micro-aggregates. The study by *Jha et al. (2012)* found that the $CO_2$ released from soil aggregates increased with increasing size class (i.e., 2–4 mm > 1–2 mm > 0.5–1 mm > 0.25–0.5 mm > less than 0.25 mm). The results of this study generally concur with these findings, however, 1–2 mm > 2–4 mm > 0.25–1 mm > less than 0.25 mm. This implies that the $CO_2$ release of soil aggregates was positive related to mass loss rate. The explanation is that there is a greater size (microbial biomass) and diversity of the microbial communities in macro-aggregates (*Jiang et al., 2011*; *Yang, Liu & Zhang, 2019*).

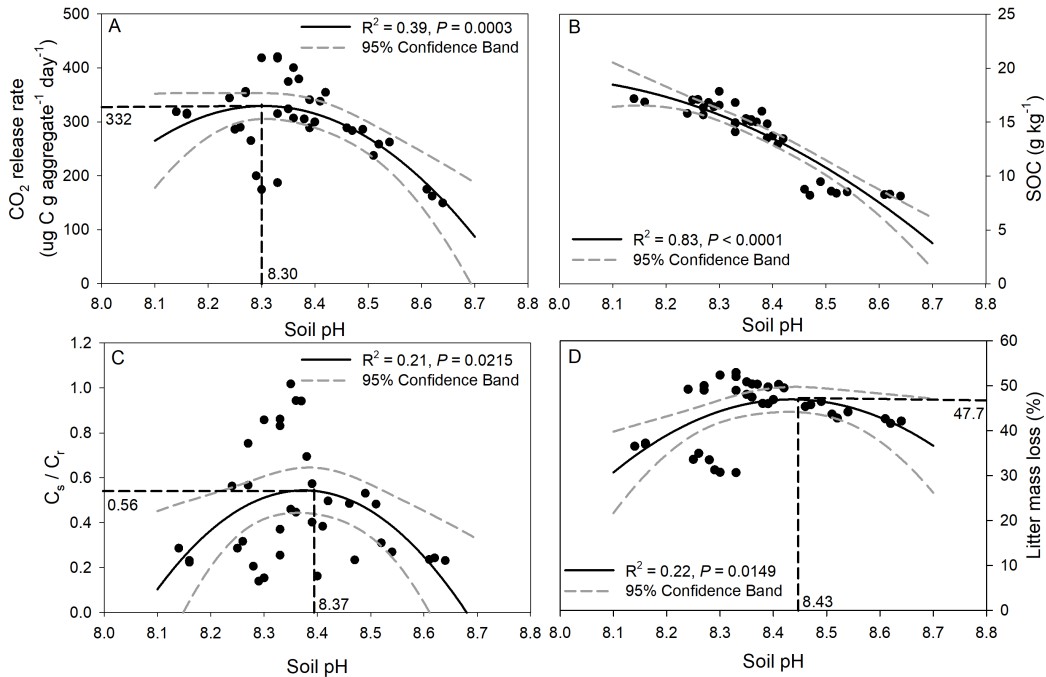

**Figure 6** Relationships between soil pH and $CO_2$ release rate (ug C g aggregate$^{-1}$ day$^{-1}$) (A), SOC concentration after incubation (B), $C_s/C_r$ ratio (C), and litter mass loss (D). The solid black line indicates a quadratic curve. The gray dashed line represents the 95% confidence interval. Black dashed line represents the vertex coordinate of the quadratic curve.

Soil aggregate size also significantly affected the SOC concentrations. It has been widely reported in many tillage systems that macro-aggregates have a higher C concentration compared to micro-aggregates (*Benbi & Senapati, 2010*; *Wang, Li & Zheng, 2017*). However, in the undisturbed grassland, we found that the <0.25 mm aggregates had higher SOC concentrations than the 2–4 mm aggregates, and 0.25–1 mm aggregates had the lowest SOC concentrations. In our research region, the soil composition of the 0.25–1 mm aggregates is mainly sand grains, which accounted for 38.26% of the bulk soil (Table 1). Hence, the low content of SOC in the smaller aggregate size class (0.25–1 mm) could be the result of lower concentrations of SOC in the 2–4 and 1–2 mm aggregates. SOC mineralization is generally higher in macro- than in micro-aggregates (*Kimura, Melling & Goh, 2012*; *Six et al., 2002*; *Tian et al., 2016*), which is in agreement with our study. The macro-aggregates, especially the 1–2 mm aggregates, had higher SOC storage than the micro-aggregates. The results from this study suggested that macro-aggregates not only have greater $CO_2$ release capacity, but also have greater litter C storage capacity than micro-aggregates.

## Effect of soil pH on carbon storage capacity of soil aggregates

The difference values of soil pH in aggregates in this study could regulate the $CO_2$ release from soil aggregates. Soil pH is a major factor that influences the structure of a soil microbial community (*Fierer & Jackson, 2006*) and thus affects the release of $CO_2$ from soil (*Andersson & Nilsson, 2001*). Some previous studies demonstrated that increases in soil pH

were highly correlated with $CO_2$ release (*Kemmitt et al., 2006*). Models indicate that SOC decomposition increases almost linearly between pH 4 and 6 (*Leifeld et al., 2013*), and *Grover et al. (2017)* suggested that liming stimulate SOC mineralization in two acidic soils. However, this study showed significant negative quadratic-correlations between the $CO_2$ release rate and the pH in the highly alkaline soils, which implied that when soil pH is greater than a certain value (8.3 in this study), the $CO_2$ release rate will decrease. The decrease in the $CO_2$ release rate at high pH may be attributed to decreased microbial biomass and activity at high pH. Our study also indicates that the SOC concentration of soil was negatively correlated with soil pH though a quadratic trend, and these results concur with those of *Kemmitt et al. (2006)*, who found a significant decline in SOC concentration with increasing soil pH. The decrease in SOC at high pH may be ascribed to the proportion of the amino acid-C taken up by the microbial biomass that was subsequently mineralized to $CO_2$ and was negatively and non-linearly correlated with pH (*Kemmitt et al., 2006*). However, *Egan, Crawley & Fornara (2018)* reported that soil pH in the acid soil was significantly positively related to greater soil C pools of smaller aggregate fractions after lime addition, implying the strong interaction of lime ($CaCO_3$) with small aggregates (clays), and *Grybos et al. (2009)* reported a positive correlation between DOC concentrations and pH in acidic soils with pH ranging from 5.5 to 7.4, and the reductive dissolution of Mn- and Fe-oxyhydroxides was the key factor controlling DOC concentrations under acidic conditions. However, those relationships reversed to negative in the alkaline soil of the present study. It should be emphasized that the pH buffering systems in acidic and alkaline soils are different, and two main pH buffering mechanisms in soils have been proposed, namely buffering by carbonates in alkaline soils with high pH (>7.5) and by aluminum compounds in acidic soils with low pH (<4.5) (*Bowman et al., 2008*; *Lieb, Darrouzet-Nardi & Bowman, 2011*). Under our research regions with higher temperature and lower precipitation, in which potential evapotranspiration greatly exceeds precipitation, carbonate tends to accumulate and thereby enhance soil pH buffering capacity in the surface soil layer (*Luo et al., 2015*), whereas in regions with higher precipitation, leaching processes prevent the accumulation of carbonate, and change the soil acidification rates. It is also interesting to note from this study that significant negative quadratic relationships were observed between soil pH and $C_s/C_r$ ratios and litter mass loss although both were relatively weak, and the respective vertex coordinates of the quadratic function were 8.37 and 8.43 for soil pH, respectively. In general, the microbial diversity associated with soil aggregates has been reported to be heterogeneously distributed (*Yang, Liu & Zhang, 2019*). The proportion of bacteria in soil varies with aggregate size, and the proportion of bacteria in micro-aggregates is larger than that in macro-aggregates (*Neumann et al., 2013*), and linking soil microbial diversity with SOC storage/release in different aggregate size classes requires further study.

## CONCLUSIONS

Our results showed that aggregate size from highly alkaline soils, litter type, and their interaction can significantly influence litter mass loss, $CO_2$ release, and SOC

concentration. The mass loss in the 1–2 mm aggregates was significantly greater than that in the other soil aggregates on both day 28 and day 56. Moreover, the 1–2 mm aggregates had the highest $CO_2$ release ($C_r$) across all treatments, while the 0.25–1 mm aggregates had the lowest. In addition, soil aggregate sizes and soil pH were correlated, and significant negative relationships were observed between soil pH and SOC concentration and $CO_2$ release. After incubation for 28 and 56 days, the SOC storage ($C_s$) was also highest in the 1–2 mm aggregates, which implied that macro-aggregates have not only a higher $CO_2$ release capacity, but also a greater litter C storage capacity than micro-aggregates. An understanding that macro-aggregates can increase SOC content has validity when trying to understand how to manage soils for increased C sequestration in the highly alkaline soils of north China. It is also important to link soil microbial abundance/diversity with SOC storage/release in different aggregate size classes, and further research in this area is needed.

## ACKNOWLEDGEMENTS

We are grateful to the workers at the National Field Research Station of Grassland Science, Guyuan, Heibei, China for their assistance during field work.

### Funding

This work was supported by the National Natural Science Foundation of China (No. 31830092). The funders had no role in study design, data collection and analysis, decision to publish, or preparation of the manuscript.

### Grant Disclosures

The following grant information was disclosed by the authors:
National Natural Science Foundation of China: 31830092.

### Competing Interests

The authors declare that they have no competing interests.

### Author Contributions

- Chao Yang conceived and designed the experiments, performed the experiments, analyzed the data, contributed reagents/materials/analysis tools, prepared figures and/or tables, approved the final draft.
- Jingjing Li performed the experiments, prepared figures and/or tables, authored or reviewed drafts of the paper, approved the final draft.
- Yingjun Zhang conceived and designed the experiments, authored or reviewed drafts of the paper, approved the final draft.

### Data Availability

   The raw data are available in the Supplemental Files.

## Supplemental Information

Supplemental information for this article can be found online at http://dx.doi.org/10.7717/peerj.7949#supplemental-information.

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
