# Peer review of "Soil aggregates indirectly influence litter carbon storage and release through soil pH in the highly alkaline soils of north China"

_PeerJ, doi:10.7717/peerj.7949_

## Round 0.1 · original submission · Major Revisions

The English needs a thorough revision, please consult a language editor or a specialist to help revised the manuscript. The reviewers are also concern about the discussion which are mainly speculative and did not address the research questions.

Reviewer 1 ·

Basic reporting

The manuscript is written quite well, easy to read and follow and with little errors. References are extensive. Figures and Tables are well presented in general, although I think that Figure 1 is very busy and the repetition of the legend is not needed and distracting. I don't think that relationships Figure 6, except maybe SOC and pH although I have issues with the small pH range, and the interpretations that can be made form this. The links to hypotheses in the discussion are somewhat speculative, and not directly linked with data (exudate quality, microbial community and range of pH is averaged across aggregate size which is confounding).

Experimental design

The experiment itself is quite detailed and well executed I think. The level of measurements and amount of data and synthesis of this is quite good and detailed. My main concerns are related to the initial aim of the study, the litter treatments (the fundamental reason for looking at the different combinations or types), their little difference in C:N (only 23.95-29.66, which would all require external N for mineralisation/decomposition) and the little range in pH (0.5 pH units), and the fact that the effects of pH are assessed by pooling aggregate size classes.

Validity of the findings

Residue type is largely ignored in the discussion but is one of the treatments.

I question the size of the litter fragments (1 cm long), the length of the incubation (56 days) and whether these are conducive to allow these to decompose sufficiently and interact with smaller aggregate size classes, which essentially biases the moderate size class (1-2 mm).

Much of the discussion is speculation and individual examples are outlined below in general comments.

Additional comments

microbial activities - this needs to be considered a single parameter, the contribution of components of the microbial community cannot be analysed separately, and as such it is one, microbial activity.

It is not clear whether leaf and stem were also dead? all are essentially standing biomass? the terms and description are confusing. is all (treatment 3) a mixture of the other 2?

Maybe it would be better to show the bar charts as values of each treatment relative to the corresponding CK? it is often difficult to see whether there has been relative effects of the treatments or these differences are all proportional to the original CK.

False wheatgrass/chinese ryegrass? the common name and relevance of the plant should be added.

The term quickly is subjective.

Add the rate (1.5% w/w or similar). First paragraph pf the M&M could be condensed.

How did you measure and maintain 90% RH?

After the stabilization period? this implies something different occurred in the second phase but essentially you did nothing different?

The start of the second paragraph repeats reps and treatments etc and is not needed.

Cleaning litter with water? didnt you lose C and N?

Soaking with HCl using auto analyser? this needs more detail on thew size, concentration and amounts of acid and materials.

Headings of M&M and results are too interpretive and should just be descriptive. The are more fitting for a discussion.

Why did you do the one-way ANOVA? you need to have some justification here? because of the large effect of litter ? but you don't mention this is the discussion?

Spaces between numbers and units (mm) are not consistent.

The first sections of the results just state significant effects and do not have good comparisons of the relative effects. The section on pH is done much better (% differences, X times more than Y etc)

CO2 accumulation, may be better as cumulative CO2?

Higher mass loss, more CO2 emissions BUT higher CO2 storage. The concentrations not easily digested and understood. Essentially this means that other treatments had less mineralisation and greater amounts of the added materials remaining as undecomposed plant fragments?

Line 52, you are speculating but you have the data?

Line 160 et al. I don't understand the acidification that has occurred in the litter-amended soils? this process is usually an alkalinising process? This needs some explanation? at this pH nitrification should not have been inhibited so the reason for a net increase in pH is unclear?

Mass loss rate is an odd term since only two time points and presented.

L184, I find it interesting that you only consider two aggregate size classes here and ignore the finer size classes. 184-186 is all speculation and not based on your data.

L199 implied. speculation.

L205. what does heterogeneous distribution of the microbial biomass mean here? this section is on soil pH? you didn't test the soil pH value difference in aggregates? (i.e. you don't have a range of pH values within each aggregate class and you didn't set up an experiments to test this). therefore your pH range is confounded by pooling the different aggregate size classes. I don't see the value in the next section (L208-210) where a correlation is derived from such a little delta in soil pH (only 0.5 pH units). All are mildly alkaline. The figure is very messy. Kemmitt had a much greater range of pH values and it is therefore not surprising they got as different relationship (L216).

What is the relevance of reducing conditions to the current study (L219).

L224, which is the most favorable for litter decomposition and soils C storage? for your data and I think that this is confounded by the fact that not a sufficient amount of time has occurred for 1-cm leaf fragments to get into these smaller size fractions.

Much of the conclusions make some bold statements from an experiment of the narrow pH range etc Similarly, suggesting that microbial community and its composition are drastically changed by the pH shifts are not valid I don't think.

If you consider the relative differences between pH are smaller than the CK's. Actually the 0.25-1mm aggregate size has had the greatest relative change in soil pH, then maybe the <0.25 and these effects are overlooked.

Reviewer 2 ·

Basic reporting

Improve the English language carefully, where needed (see some comments below)

Experimental design

The experimental design is okay and research questions are well defined.

Validity of the findings

Novelty is also okay; the conclusion part needs some improvement such as linking the messages to the key research questions.

Additional comments

Reviewing Manuscript 38613v1

This is an interesting study where the authors examined the role of soil aggregate size classes and their interactions with different types of litter on carbon storage capacity in soil. The manuscript is generally well written and the listed questions in ‘Introduction’ are well explained (as per the findings of this study), but there is a need for further improvements of the whole manuscript. The authors need to make the right use of articles, plus to also fix a few grammar issues. There are a few issues with some figures, which should be clarified/improved in a better way. All (sub)-sections and many figures of this manuscript are mostly okay but the authors need to address some of the issues before the publication of this manuscript in PeerJ (see some specific comments below).

Specific comments:

Improve the title of this manuscript: for example, clarify the meaning of “through soil pH”.

Abstract

Line 11: The authors mentioned microbial activities but these were not examined.
Line 13: Change ‘is’ to ‘are?
Line 17: Clarify that this is the “Litter mass loss rate”.
Line 17: The authors should present the data and statistical outcomes of Cr and Cs in relevant figures; see below.
Lines 24-25: What would be the broader implications of these results (CO2 release and litter C storage) from the aggregate size classes (macro- vs. micro-aggregates), while considering these results are from one soil type only; clarify.


Introduction

Lines 44-46 and then Line 48: Yes, this is important with growing evidence, but why the authors are talking about “decomposer community composition”; their research results did not present any data on microbial activities/community composition, etc.

Lines 56-57: What do the authors mean by “higher conversion capacity”? Clarify.


Material and Methods

Line 77: Change ‘dry sieve’ to ‘dry sieving’.

Line 82: As the authors may know, there would be some reasonable quantities of silt (<0.053 mm) and clay (< 0.002 mm) particles in less than <0.25 mm micro-aggregates, which would still not be aggregated as micro-aggregates (<0.25 mm) and micro-structures (<0.053mm). These data need to be evaluated and reported, or at least to say that “micro-aggregates (<0.25 mm)” will also have silt-plus-clay particles and micro-structures. Then after clarifying this, they can classify all these size classes as “micro-aggregates” for reporting in this study. Look at some relevant published studies and cite those.

Lines 97: Hope the authors would have added the plant litter in each soil aggregate after the pre-incubation period for three days. This is a usual approach and is important. Clarify.

Lines 103-104: Improve the message here.

Line 119: Is there any reference where previous incubation studies may have used this carbon storage capacity approach (i.e. Cs/Cr) over only two months? Cite a couple of those studies.

Lines 125-126: Why this statistical approach (one-way ANOVA) was also done? Clarify or remove this sentence. This best approach is the two-way ANOVA, which is already mentioned above (Line 124).


Results

Lines 142-144: Clarify/explain the significant differences for the other aggregates size classes also.

Lines 148-150: There are a few issues. Fix citations of the right Figures for different statements; remove the irrelevant ones. The data presented in Figure 3 are just the SOC concentrations but not the storage; hope the authors can also present the SOC storage data, which is Cs (SOC in litter treatment minus SOC in the control treatment). Was the SOC storage really high in 1-2 mm aggregates? If the authors can also present the Cs data in relevant subfigures, then it will be clearer for the readers. There is also a minor grammar issue here (and also a few other places; carefully improve the manuscript while fixing similar English grammar issues).

Line 152: Is the impact of soil aggregate size and litter type interaction really significant on the Cs / Cr ratio? This is not right as per the statistical results presented in Table 2 for the Cs / Cr.

Lines 160-161: This information does not seem to be provided “significantly lower mean soil pH”?

Lines 168-170: Fix any minor English grammar/language issue.



Discussion

Line 174: Change to “Soil aggregates regulate”

Lines 185-187: Improve the flow of this message here (Seems the sentence is not complete and not clear) so the readers can understand.

Line 188: Why saying “affected”

Line 193: Did the authors examine and report sand, silt, and clay contents in each aggregate size class; this information will be useful.

Lines 199-200: Clarify this information here, and in Figures 2 and 3; the results presented here are just the litter treatments and controls but the data and statistics are not presented for Cr and Cs values (for example, after subtracting the control SOC from litter SOC and then litter C storage (Cs) and their statistical results will be clearer for the readers.

Lines 204-205: Clarify and improve this statement/message. Then fix some minor English issue (e.g. at Line 205).

Lines 214-216: Improve the message and the English language flow here. What do you mean by “both soils”; this study only looked at one soil but then different aggregate size classes.

Lines 225-226: This study is about aggregate size classes and the information on microbial structure and diversity in soil aggregate size classes needs further studies, although some work has been done; carefully look at the literature and improve the flow of the message here and cite those new studies.



Conclusions

I feel the conclusion section would only briefly mention the key results but the messages need to be linked well the questions/hypotheses developed in this manuscript, for example, in the Introduction section.


Line 229: clarify that the mass loss was related to “litter mass loss”.

Line 231: Change CO2 accumulation to CO2 release.

Lines 236-237: Mention implications of these future work for SOC storage; of course the microbial diversity will change with changing pH, but it is important to link soil microbial abundance/diversity with SOC storage/release at different aggregate size classes, and more of this work is needed in future research activities. Clarify this information for future work.

Table 1: Did the authors also analyze inorganic C content in all aggregates?

Figure 2: Cumulative CO2 data for both litter treatment and the control are also presented in Figure 1 (g, h, i), including for days 28 and 56. It would be better to also present Cr (CO2 from litter minus CO2-C from control) and their statistical comparisons. Ditto for Figure 3 (Cs = SOC-L minus SOC-CK); present these data also (i.e. Cs).

Reviewer 3 ·

Basic reporting

The article has some issues with clarity and language. Please see my points below:

Please see general comments for all Review points.

Experimental design

I think the experimental design is ok, however, I have some concerns about drying soil aggregates as this may effect microbial composition, abundance and activity. This may be a standard method, however, can you please provide a suitable reference or further explanation.

One major component of the experiment is to measure how much plant litter is incorporated into each aggregate fraction microcosm. This goal needs to be made clearer. To do this please replace 'mass loss' with 'litter mass loss' or 'litter mass incorporated'.

Validity of the findings

The main findings are that macroaggregates (1-2mm) incorporated that greatest amount of leaf litter (i.e. litter mass loss was greatest) and the macroaggregates had the highest CO2 accumulation and release. These findings need to be made clearer in the conclusions. Specifically, the author needs to correct the error where they say '. . . 2mm aggregates had the highest CO2 accumulation (Cr)'. This is an error and contradictory as Cr is C released. Instead the author needs to write something like, 'Moreover, the 1-2mm aggregates had the higher CO2 accumulation (Cs) and loss (Cr) for all treatments, while the 0.25-1mm had the lowest'. That is if this is true also for the 0.25-1mm.

The author also needs to state in the conclusion that C storage in response to leaf litter is greater than C released within the soil macroaggregates and point the reader to the graph (Fig. 4) which shows the ratio of Cs/Cr.

An understanding that macroaggregates can increase SOC content has validity when trying to understand how to manage soils for increased C sequestration. A similar conclusion needs to be made in the paper.

Additional comments

Review of paper: Soil aggregates influence litter carbon storage and release through soil pH

1. Line 13. The Background section of the Abstract does not provide enough context to allow the reader to understand what the paper is going to be about. My suggestion is to add a sentence after that says something like, ‘Here we ask how four different aggregate sizes influence rates of C release (Cr) and SOC storage (Cs) in response to three types of plant litter added to an ungrazed natural grassland. The remainder of the abstract needs to be amended as appropriate to avoid duplication.

2. Lines 19 – 25. The Results section should be written more simply to focus on the main messages in relation to the hypotheses set. This should say something like: (i) A significant negative relationship was found between Carbon stored (Cs) / Carbon released (Cr) and soil pH, and (ii) Macro Aggregates (1-2mm) had the highest CO2 release and accumulation rates of all treatments.

3. Line 39. After low lignin content, consider adding, (i.e. low C-to-Nut ratio).

4. Line 46. Remove the word ‘controls’. I think the sentence can end at litter quality.

5. Line 46. I don’t understand the sentence, ‘Soil microbial processes are regulated by constraints in soil pH . . . ‘. Please explain further. It is true that soil pH is a main factor regulating soil processes, including plant and microbial activity and composition with effects on soil aggregate stability and development.

6. Line 50. I don’t like the sentence, ‘Soil aggregates are structural units of physical barriers . . . . . ‘. I think it should be worded something like, ‘Soil aggregates provide physical protection of contained SOC and nutrients which can change the rate of C and nutrient cycling’.

7. Line 53. I think an explanation needs to be added at the end of the following sentence, ‘SOC mineralization in macro-aggregates is considered to be higher than micro-aggregates (Fernandez et al. 2010; Rabbi et al. 2014), because . . . . ‘. Please see Six et al. (2004) (already in your reference list or Stamati et al. (20013) for an explanation. Stamati’s paper is titled, ‘A coupled carbon, aggregation, and structure turnover (CAST) model for top soils’.

8. Line 64. Change to something like: ‘(i) soil aggregate size and pH are correlated and effect rates of litter decomposition’. Delete consequently. For hypothesis (ii) change to something like, ‘macro aggregates will have higher litter C concentrations than micro aggregates, despite greater losses of carbon.

9. Line 74. Change to the soils were spread.

10. Line 76. Change the sentence to something like, ‘Soil aggregates were separated into different size fractions by dry sieving in accordance with the method in Elliott (1986).

11. Line 86. Change Ca. to Ca. (Italics). Acceptable without so up to you.

12. Line 93. Change to ‘. . . was used as a control (CK).

13. Line 141. The sentence is written for an introduction and not for the results section. Please change to something like, ‘Soil aggregate size, litter type and their interaction has significantly influenced mass loss, CO2 release and SOC concentration.

14. Line 144. Change to ‘In all three litter addition treatments, . . . .’

15. Line 146. Change to, ‘. . . . . remaining decomposition for all the four aggregate sizes. The ‘s’ needs to be added.

16. Line 150. Add the words, ‘the’ and ‘fraction’ to the sentence so it reads, ‘. . . . was highest in the 1-2mm aggregate fraction.

17. Line 178. Change the sentence to read, ‘. . . significantly higher rates of loss occurred in soil . . . . ‘.
18. Line 179. Change the sentence to rear, ‘This may be because of greater contact with air filled pores and microorganisms . . . . ‘.

19. Line 180. Consider changing the sentence that begins on line 180 to read as follows, ‘The study by Jha found that the CO2 released from soil aggregates increased with increasing size class (i.e. 2-4mm > 1-2mm > 0.5-1mm > less than 0.25 mm; Jha et al. 2012). The results of this study generally concur with these findings, however, macro-aggregates (1 – 2mm) released more CO2 than large macro aggregates (2-4mm).

20. Line 192. Add the word, ‘the’ to the following sentence, ‘ . . . . . aggregates has the lowest SOC concentrations.

21. Line 194. Add the word, ‘the’ to the following sentence, ‘Hence, the low content of SOC in the smaller aggregate size class . . . ‘.

22. Line 195. Remove the end of the same sentence. Remove, ‘than <0.25 mm aggregates’. I don’t think this is required.

23. Line 232. Add the word, ‘negative’ to the following sentence, ‘In addition, significantly negative relationships were . . . . ‘.

24. If possible, carry out a mass balance to understand if rates of CO2 release in macro aggregates are > or < then rates of C storage.

25. The results of this paper show a negative relationship between pH and SOC etc. This is contrary to many papers that show a positive liming effect on C and N pools e.g. Egan et al. (2018), ‘Effects of long-term grassland management on the carbon and nitrogen pools of different soil aggregate fractions’. Could the difference be mineralogy and interaction of lime (CaCO3) with clays?? This needs to be included in the discussion.

Tables and Figures

T. 1. Add the word initial or similar so the reader knows what data this table relates to, ‘i.e. Initial mean . . . ‘
T.2. I do not understand this table. The description says it is aggregate size and litter type v. the other variables (Mass loss, CO2 release rates, SOC, Cs/Cr). If so, why is there only one size and one type?

Shouldn’t the graph look like the attached PDF

Overall I think it is a good paper. However, the authors need to decide what are there key messages and structure the paper accordingly.

Annotated reviews are not available for download in order to protect the identity of reviewers who chose to remain anonymous.

---

## Round 0.2 · Major Revisions

The paper has been thoroughly revised. However there are still few snags that need to be improved. In particular, the title, abstract, and content of the paper need to mention that this experiment is on
calcicorthic Aridisol.

The soil used in this experiment is highly alkaline (8-9) with CaCO3, and thus the results are only applicable in this type of soil. I would suggest modifying the title to reflect the soil type. There is no indication of the soil properties, and that needs to be included in the manuscript, sand, clay, pH, CEC, exch. cations, SOC, CaCO3, Gypsum.. Also include the temperature and precipitation regime of the area where the soil comes from.

Would the CaCO3 or CaSO4 in the soil influence the decomposition?
The abstract also needs to mention the soil type used in this study (highly alkaline soil).

The soil's pH buffering system in this lakaline condition is different from other soils, this needs to be discussed.

Reviewer 1 ·

Basic reporting

All fine. Thorough revision. Think that the manuscript is acceptable for publication with some minor revision.

Experimental design

All fine. Thorough revision. Think that the manuscript is acceptable for publication with some minor revision.

Validity of the findings

All fine. Thorough revision. Think that the manuscript is acceptable for publication with some minor revision.

Additional comments

Thorough revision. Think that the manuscript is acceptable for publication with some minor revision.

Comments/suggestions.

L40. decomposer activity? Composition and activity of the soil decomposer community?

L53. The word contained is not needed I don’t think.

L57 maybe higher should be greater? (mineralization)

L58 a reduced microbial activity? ‘a’ is not needed.

L63. Litter decomposition capacity of carbon ... needs rewording.

L101 chemical properties?

L111 before adding litter?

L113 pre incubation period?

L122 shaking soil with water (1 :2.5 w/v) for 30 min.

L129 Burette?

L136 please insert a space between 1 and indicates.

L138 zone? Not clear what you refer to here.

L148 chemistries? Better to say that the initial chemical composition of all three litters differed substantially. Place a space between 1 and Table.

L156 C/N ratios were higher?

L166 then a decrease is misleading, since the rate slowed from the initial rapid rate but didn’t in-fact decrease

L171 can’t have concentrations (plural), with the and was (singular)

L174 too many spaces between pH and carbon

L175 can is not the right word here as it is speculative, but you are using actual data.

L176 has ‘the’ highest

L183 decreased the pH in the soils than the same soils? This suggests that the drop in soil pH was the same with and without litter? Then this is not an effect of the litter? But the data show that the drop in pH is the same for all aggregate class sizes?

L201 chemical composition of the litters?

L203 decay not decays.

L206 compared with.

L209 the use of the refence here suggests that the statement is based on their data (Jha) and not yours also, it needs a conjunction I think as the statement is for your data but also shown by Jha? Have I interpreted this correctly?

L216 The explanation is that there is a greater size (microbial biomass) and diversity of the microbial communities in macro-aggregates?

L218 significantly affected SOC concentrations

L233 The different values of soil pH

L237-247 this whole section about soil pH and SOC mineralization only refers to one reference but there are others that have explored this phenomenon. E.g. Grover, S.P., Butterly, C.R., Wang, X. et al. Biol Fertil Soils (2017) 53: 431. https://doi.org/10.1007/s00374-017-1196-y

L254 significant negative quadratic relationships? Also change in other places.

L265 greater instead of higher?

L268 were correlated? Significant negative relationships?

L271 greater litter C storage capacity

L275 in different aggregate size classes? And further research in this area is needed?


I still feel that the repetition of the legend on Fig 1. Makes it too busy and they are not needed on every single panel. The Editor should advise on this.

---

## Round 0.3 · accepted · Accept

The authors have addressed most of the reviewer's concern and particularly define the highly alkaline soil used in this study